# Nanoparticle-enabled phase control for arc welding of unweldable aluminum alloy 7075

Maximilian Sokoluk[1], Chezheng Cao[2], Shuaihang Pan [1] & Xiaochun Li[1,2]

Lightweight materials are of paramount importance to reduce energy consumption and emissions in today's society. For materials to qualify for widespread use in lightweight structural assembly, they must be weldable or joinable, which has been a long-standing issue for high strength aluminum alloys, such as 7075 (AA7075) due to their hot crack susceptibility during fusion welding. Here, we show that AA7075 can be safely arc welded without hot cracks by introducing nanoparticle-enabled phase control during welding. Joints welded with an AA7075 filler rod containing TiC nanoparticles not only exhibit fine globular grains and a modified secondary phase, both which intrinsically eliminate the materials hot crack susceptibility, but moreover show exceptional tensile strength in both as-welded and post-weld heat-treated conditions. This rather simple twist to the filler material of a fusion weld could be generally applied to a wide range of hot crack susceptible materials.

[1] Department of Mechanical and Aerospace Engineering, University of California, Los Angeles, CA 90095, USA. [2] Department of Materials Science and Engineering, University of California, Los Angeles, CA 90095, USA. Correspondence and requests for materials should be addressed to X.L. (email: xcli@seas.ucla.edu)

Today, lighter materials and/or structures are crucial for reducing fuel consumption and emissions for all transportation applications[1], especially for automotive and aerospace industries. It has been indicated that a 10% weight reduction results in a 6–8% increase in fuel economy for automobiles[2]. Assembling lightweight structural parts to a functional assembly is thus of paramount importance for today's society and industries such as modern automotive, construction, shipbuilding and aviation. Welding or joining requires minimum effort and cost for effective design and therefore weldability is widely considered as a central aspect in the process of qualifying materials for widespread use.

The use of lightweight aluminum (Al) in vehicles has been increasing rapidly throughout the last decade. By using Al alloys of higher strength, the vehicle weight can be further reduced. With their superior high strength to weight ratio, heat-treatable Al alloys, especially the 2xxx and 7xxx series, often find applications in today's aerospace or military industries[3,4]. The most widely known specimen of 7xxx alloys is Al alloy 7075 (AA7075), an Al-Zn-Mg-Cu alloy, which offers exceptional strength at low weight. However, these materials are notorious for their susceptibility to cracking during arc welding, thus significantly limiting their widespread use. Due to the unweldability of AA7075, and high strength aluminum alloys in general, currently the aerospace industry strongly relies on riveting and bolting to join these materials to a complex assembly. Recently friction stir welding (FSW), a solid state joining technology, has been successfully implemented to join AA7075. However, the fully mechanized nature of the FSW process prevents its use for applications where access or weld shape is complicated. The workpiece also needs to be restrained in well-designed support tooling, making complicated welds in FSW difficult to set up. Therefore, while arc welding of AA7075 is still highly desired for airplanes or vehicles, its arc weldability remains as a barrier.

A significant amount of research has been conducted in order to enable arc welding for high strength aluminum alloys, especially AA7075[5–16]. These approaches, to optimize welding parameters or find metallurgic remedies for the unweldability of these materials, unfortunately could not solve the long-standing problem. Lately a new approach has been made to introduce nanotechnology to solidification processing, such as casting and arc welding. Nanoparticles are not only known to enhance the properties of metal matrices; they also strongly influence grain growth and solid fraction of alloys during solidifying, along with the viscosity and thermal properties of the melt[17–19]. Choi et al. found that nanoparticle enhancement can serve as a remedy for hot tearing in the hot tear susceptible aluminum cast alloy A206 (Al-4.5Cu-0.25Mg)[17]. To enhance the mechanical properties of the melt zone material in arc welding, filler materials containing nanoreinforcements have been tested[20–25]. However, little success regarding the arc welding of the traditionally un-weldable AA7075 has been reported.

Here we show that a nanoparticle-enhanced AA7075 filler rod can be used to weld the notoriously unweldable material. This approach intrinsically eliminates the occurrence of solidification and liquation cracking, the two dominant hot crack modes that have kept the arc welding of AA7075 from being successful. Furthermore, the resulting welds offer grain sizes of about 9 μm in the melting zone without diluting the strengthening elements in the melting zone. The welds showed a ultimate tensile strength of up to 392 MPa in as-welded condition, while reaching up to 551 MPa with post weld heat treatment (PWHT). This suggests that the introduction of a nanotechnology-treated welding rod to arc welding of AA7075 enables fusion welding for structural design of this high-performance alloy for mainstream applications such as electrical vehicles. This nanotechnology-treating approach can be readily extended to welding of other hot crack susceptible materials beyond the 2xxx and 7xxx series aluminum alloys.

## Results

**Arc welding results**. The nanotechnology-treated AA7075 filler rod was fabricated by incorporating 1.7 vol% of 40–60 nm TiC nanoparticles into AA7075 using salt assisted nanoparticle incorporation and hot extrusion (see Methods). A major advantage of this fabrication process over particle coated filler rods or filler tubes with a nanoparticle core is the state in which nanoparticles are introduced into the melt pool while welding. By already incorporating dispersed nanoparticles into an aluminum matrix prior to the actual welding process, the transition of reinforcements into the melting zone is more effective. For the purpose of comparison, we conducted arc welding experiments using standard AA5356 (Al-5Mg) filler, pure AA7075 filler and our nanotechnology-treated AA7075 rod to fuse two AA7075 sheets with a dimension of 152.4 × 76.2 × 3.175 mm each, as shown in Fig. 1a (see Methods). Welds performed with these three different filler materials are shown in Fig. 1b–d. It

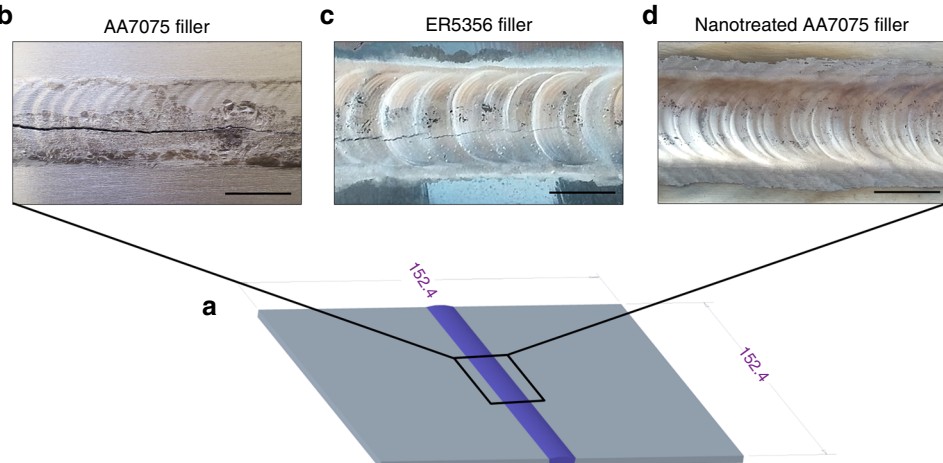

**Fig. 1** Gas tungsten arc welding (GTAW) of AA7075. **a** Two 152.4 × 76.2 × 3.175 mm AA7075 sheets were arc welded using three different types of filler materials for the weld bead (purple). **b** and **c** Macroscopic solidification cracks in the bead's melting zones in the welds performed with conventional filler materials AA7075 and ER5356, respectively. **d** Using AA7075 + 1.7 vol% TiC as filler material, the weld yields an even weld bead without signs of cracking. Scale bars, 10 mm

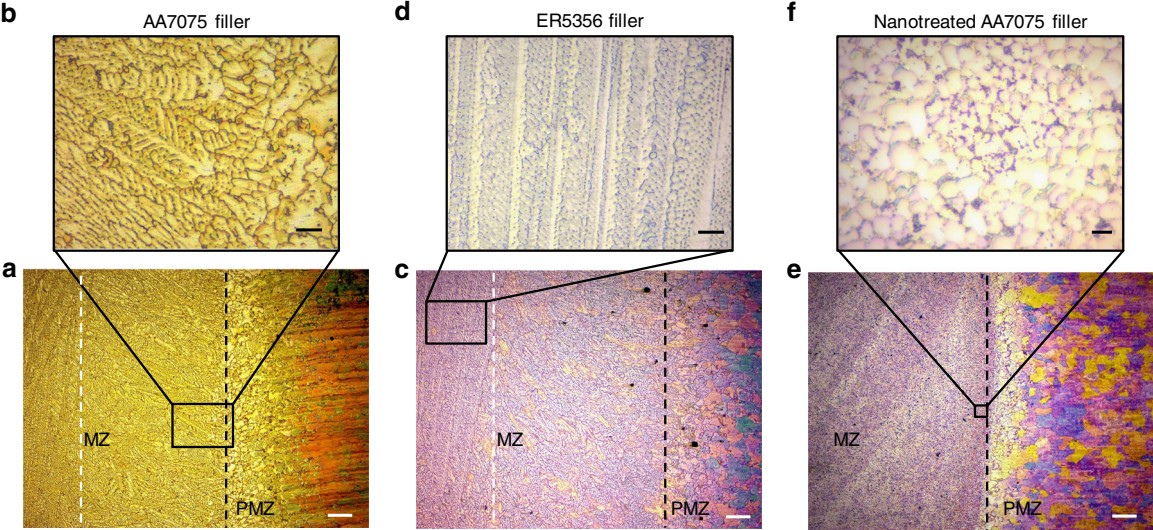

**Fig. 2** Optical microscope characterization of horizontal weld bead cross-sections, in-plane with the welded base plates, as shown in Fig. 1. **a**, **c** and **e** Grain morphology of welds performed with AA7075, ER5356 and nano-treated AA7075 as filler material, respectively. The black dashed lines indicate the fusion line of melting zone (MZ) and partially melted zone (PMZ), while the white dashed lines indicate the transition of curved grain growth adjacent to the fusion line and columnar, trailing grain growth in the weld bead's center. **b** Dendrite originated from fusion line. **d** Columnar grains in the weld's center section. **f** Globular grains in MZ of nano-treated AA7075 weld. Scale bar, 200 μm in **a**, **c**, **e**, 50 μm in **b**, **d**, 10 μm in **f**

should be noted that the weld parameters, with relatively high heat input and relatively slow welding speed, were chosen to increase the thermal stress, as well as the solidification shrinkage and therefore showcase the filler materials' impact on solidification crack susceptibility.

Figure 1b shows a weld performed with pure AA7075 filler material. The continuous, macroscopic crack at the center line of the joint is indicative of AA7075's susceptibility to solidification cracking when fusion welded. The weld performed using ER5356 is shown in Fig. 1c. This type of filler is commonly used to mitigate the occurrence of solidification cracking when welding less crack susceptible materials, such as AA6061 and AA7005. Nevertheless, in our experiment, the effect of this filler material was unsatisfactory and did not suppress solidification cracking in AA7075. The joint showed the same error pattern as in Fig. 1b. The third weld performed with nano-treated AA7075 filler material showed a surprisingly different picture. The bead shown in Fig. 1d was free of macroscopic imperfections, while welded under the same parameters as the two conventional filler materials.

**Nanoparticle effects on grain morphology in the melting zone.** To further understand these results, we performed microstructure studies using optical microscopy (OM) and Weck's Reagent to unveil the different melting zones' grain morphologies. In Fig. 2a the drawbacks of welding AA7075 with similar filler material become obvious.

Due to the alloy's wide semi-solid zone and non-linear solid fraction vs. temperature curve, the melting zone solidifies in large dendrites trailing the heat input. Inset Fig. 2b shows curved dendritic grains adjacent to the fusion line with an average size of $116.5 \pm 68.7$ μm, while closer to the centerline the highly dendritic grains reach a length of hundreds of micrometers. During solidification these dendrites form liquid trenches that must be supplied with liquid from the MZ while gradually transitioning from liquid to solid state. If these trenches, by spontaneous nucleation at the entrance or merging of solidification fronts, are cut off from the MZ the remaining liquid fraction forms a pocket. The volume shrinkage of the trapped liquid will cause perpendicular tension regarding the welding direction. If the

pressure inside this trench drops below the cavitation pressure, a void forms, initiating a crack which trails the MZ. It was found that this phenomenon strongly relies on the solidification speed of the manufacturing process[26].

Figure 2c shows the melting zone welded with an ER5356 filler rod. It can be found that the grain size adjacent to the fusion line is reduced to an average of $70.5 \pm 44.5$ μm, while the columnar dendritic grain growth, shown in inset Fig. 2d remains similar to the weld performed with AA7075 filler material. The aim of welding with dissimilar filler materials is to heavily dilute the crack sensitive base material with less crack sensitive filler material. However, with this approach several compromises must be made. By diluting the melting zone with dissimilar filler alloys, the concentration of strengthening alloy components of AA7075 decreases. This leads to a decrease in as-weld strength, as well as a reduced responsiveness to post weld heat treatments. Furthermore, the use of these filler rods will increase the likelihood of liquation cracking. Kou et al. found that, conventional aluminum filler alloys, such as ER5356, cause the melting zone to solidify earlier than the partially melted zone adjacent to the melt pool. To be more precise, the solid fraction of the melt pool composition becomes larger than the solid fraction of pure AA7075 in the partially melted zone at the same temperature. This causes tension on the weakened PMZ towards the center of the melt pool and ultimately leads to liquation cracks[27]. This failure mode was not observed in this study due to stress relief brought about by solidification cracking.

With our nano-treated filler material, an alternative approach that has the potential of fundamentally changing the material's solidification mechanisms has been implemented successfully, enabling fusion welding for AA7075 without any of the aforementioned drawbacks. Figure 2e shows the horizontal cross-section of the joint which was welded using AA7075 treated with 1.7 vol% TiC filler rod. Figure 2e reveals a homogeneous grain morphology throughout the MZ, very different from the previously introduced conventional welds. The grains are highly equiaxed with an average size of $9.4 \pm 5.0$ μm, showing smooth grain boundaries. With the addition of nanoparticles however, the epitaxial nucleation at the liquid-solid interface remains favorable. Several beneficial mechanisms are

introduced to the solidification process causing this particular grain morphology. Firstly, the presence of TiC decelerates the solidification front and therefore reduces the speed of dendritic grain growth originating from the liquid-solid interface. The decelerated grain growth enables the formation of an area adjacent to the interface with increased undercooling, where the presence of nanoparticles adds heterogeneous nucleation to the solidification process. This effect is promoted for aluminum and TiC having a lattice mismatch factor of 5.8% which indicates complete epitaxy of the aluminum grain nucleating at the nanoparticles' surface[28]. Lastly, nanoparticles alter the appearance of these heterogeneously nucleated grains. As recently shown by Guo et al., nanoparticles have a strong impact on equiaxed dendritic grain growth[19]. The authors observed that SiC nanoparticles, modified the structure of α-grains in a Mg-25Zn-7Al alloy, yielding nearly globular grains, similar to the grain appearance shown in inset Fig. 2f. This absence of directional, dendritic grain growth is an important indicator for the advantages nano-treated filler materials bring to the welding process of hot-crack susceptible aluminum alloys. Furthermore, the use of similar filler material intrinsically eliminates the occurrence of liquation cracking.

**Modification of the secondary phase of AA7075**. To further understand the effects of TiC nanoparticles on AA7075's solidification behavior, Scanning Electron Microscopy (SEM) and Transmission Electron Microscopy (TEM) were utilized to investigate the secondary phase morphology of melting zones welded with pure AA7075 and nano-treated AA7075 filler. Figure 3a shows the typical secondary phase distribution of a pure AA7075 melting zone. As observed in the OM images, the appearance of secondary phase in the pure aluminum alloy indicates dendritic solidification of the α-aluminum grains. At the weld center, this solidification mode leads to relatively long and continuous segregations at the grain boundaries in the welding direction.

As stated earlier, dendritic grain growth is undesired since it may cause macroscopic or microscopic cavities and cracks. Moreover, such continuous longitudinal secondary phase features cause the MZ to have low tensile strength perpendicular to the welding direction. At larger magnification in Fig. 3c, it is observed that the secondary phase of pure alloy solidifies in large eutectic Mg(Zn,Cu, Al)$_2$ areas especially at triple junctions of grain boundaries. Figure 3b shows typical secondary phase under the presence of TiC nanoparticles. Here, the secondary phase is segmented, while its fragments are randomly oriented, finer and shorter than its counterparts in pure alloy. Figure 3d shows areas of locally larger intermetallics in the nano-treated melting zone. Here, the advancing grain boundaries accumulated TiC in the terminal stages of solidification, resulting in a locally higher volume percentage of nanoparticles in the secondary phase. Therefore, the occurrence and size of eutectic features were drastically reduced. Inset Fig. 3e shows one of the few lamellar Mg(Zn,Cu,Al)$_2$ phases formed during solidification. The nanoparticles modified its regular lamellar pattern, by divorcing links within the structure. These findings are in accordance with the melting zone's reduced crack susceptibility. The finer, randomly oriented segregations and the reduced size of divorced eutectic features, in combination with round and equiaxed grains, suggest high flowability of secondary phase up to the terminal stage of solidification. Therefore, the entrapment of liquid secondary phase pockets, which are responsible for solidification cracking, is unlikely.

To clearly reveal the interface between the TiC nanoparticles and secondary Mg(Zn,Cu,Al)$_2$ phases, TEM analysis at atomic scale was utilized. Figure 3f shows an SEM image of the TEM

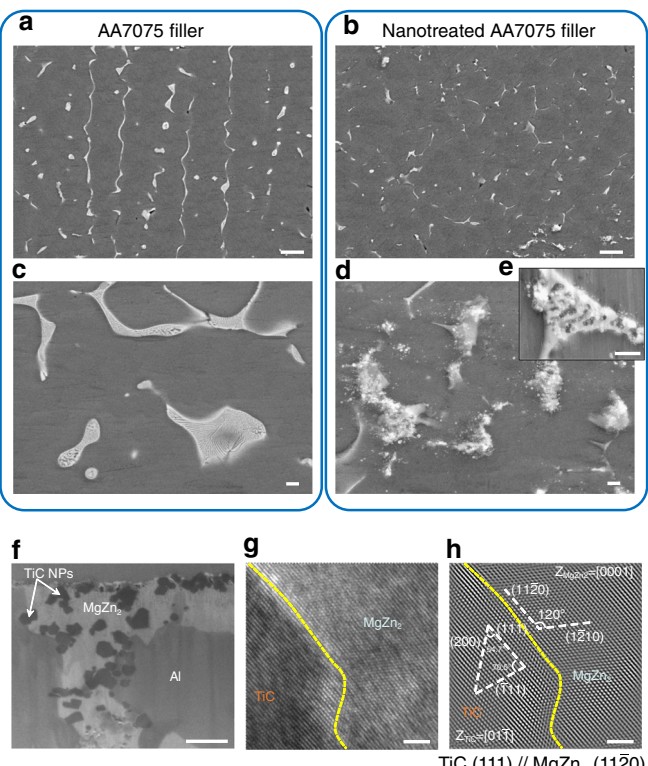

**Fig. 3** SEM characterization of horizontal weld bead cross-sections presented in Fig. 2 of pure AA7075 and nano-treated AA7075 melting zones. **a** and **c** Secondary phase microstructure of a melting zone welded with AA7075 filler material. **b**, **d** and **e** Modified secondary phase of AA7075, welded with nano-treated filler material. **f** TEM sample showing the cross-section view of the modified secondary phase by nano-treated filler material. **g** A typical high resolution TEM image of the interface between TiC nanoparticle and secondary phase (identified as MgZn$_2$). **h** Fourier-filtered high resolution TEM image corresponding to **g**. Scale bars, 10 μm in **a**, **b**, 1 μm in **c**, **d**, **e**, **f**, and 2 nm in **g**, **h**

sample cut from nano-treated eutectic Mg(Zn,Cu,Al)$_2$ areas containing TiC nanoparticles. As marked in the figure, it is observed that most TiC nanoparticles stay either inside the secondary phase or at the boundary between Al matrix and Mg (Zn,Cu,Al)$_2$ phase. This indicates that TiC nanoparticles prefer to stay within Mg(Zn,Cu,Al)$_2$ phases during solidification and effectively modify the size, shape, and distribution of secondary phases in the MZ. Figure 3g shows a typical interface between TiC nanoparticles and secondary phase. Figure 3h is the Fourier-filtered atomic resolution TEM image corresponding to Fig. 3g. The observed secondary phase was identified as MgZn$_2$ phase by its atomic structure. As marked in Fig. 3h, (11$\bar{2}$0) and (1$\bar{2}$10) planes of MgZn$_2$ were identified with an angle of 120°. The MgZn$_2$ phase is oriented towards the [0001] zone axis. Furthermore, a TiC nanoparticle's (111), ($\bar{1}$11), and (200) planes are identified and marked in the atomic structure. The observed TiC nanoparticle is oriented towards the [01$\bar{1}$] zone axis. The (111) planes of TiC are shown to be parallel with the (11$\bar{2}$0) planes of MgZn$_2$. The ($\bar{1}$11) planes of TiC have an angle of approximately 10° between the (1$\bar{2}$10) plane of MgZn$_2$. The planar distances of ($\bar{1}$11) TiC and (1$\bar{2}$10) MgZn$_2$ are 0.2499 nm and 0.2609 nm, respectively. Thus, the misfit at the TiC-MgZn$_2$ interface is calculated to be approximately 5.6%, which indicates a semi-coherent interface. The good lattice matching also explains why TiC nanoparticles tend to affiliate with the secondary MgZn$_2$ phase and effectively modify MgZn$_2$ for improved welding quality.

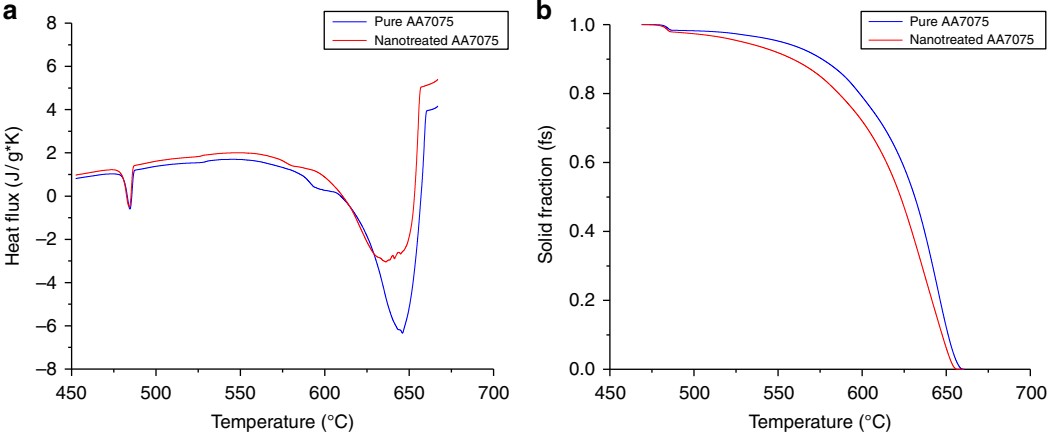

**Fig. 4** DSC analyses of pure AA7075 and nano-treated AA7075 melting zones. **a** DSC cooling curves showing the normalized heat flux of melting zone samples taken from joints welded with AA7075 and with nano-treated AA7075 filler material. **b** Solid fraction vs. temperature curves derived from DSC results

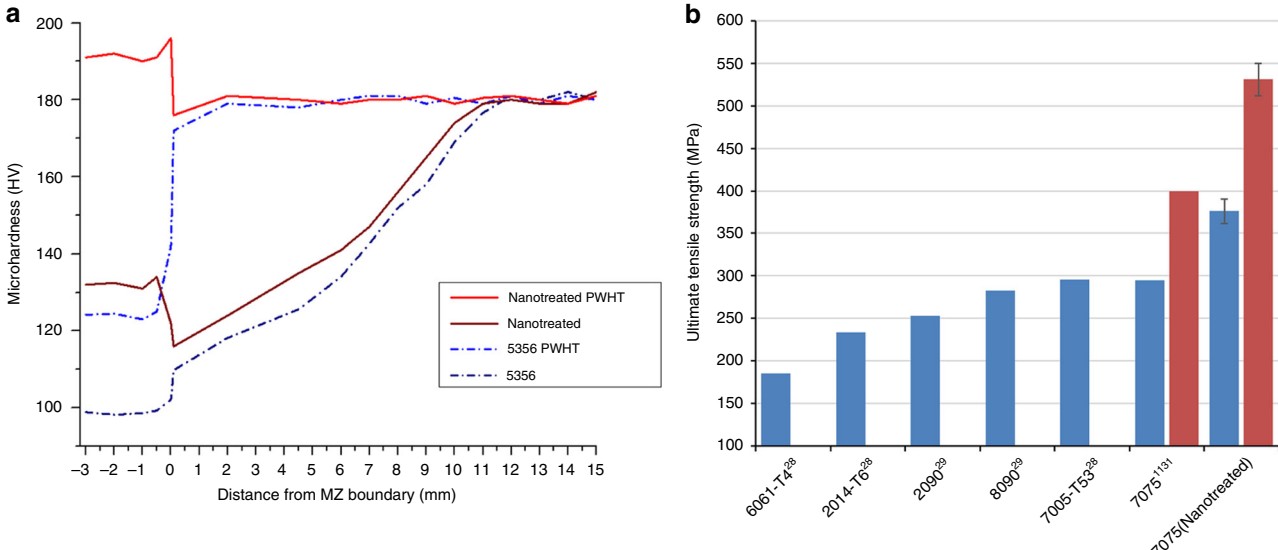

**Fig. 5** Mechanical properties of nano-treated AA7075 welds. **a** Microhardness tests were conducted at the centerline of transversal vertical cross-sections. The graph compares the Vicker's hardness values of welds performed with ER5356 filler and nano-treated AA7075 filler in as-welded and post weld heat treatment condition. **b** from left to right, the blue bars show a selection of commonly GTAW welded high strength aluminum alloys[36, 37], AA7075 welded with ER5356 filler material[11] and AA7075 welded with nano-treated AA7075 filler material. The red bars show an AA7075 modified indirect electric arc weld (MIEA) with ER5356 as filler[38] and AA7075 welded with nano-treated AA7075 filler material after the same PWHT. Error bars show s.d. of 4 tested samples

**Thermal analysis of solidification process**. To better interpret the changes TiC nanoparticles introduce to AA7075's α-grain and secondary phase morphologies, melting zone samples of joints welded with pure and nano-treated AA7075 filler rod were analyzed using differential scanning calorimetry (DSC). When comparing the two cooling curves in Fig. 4a the heat release peaks of α-grain nucleation differ significantly. The curve representing the pure material shows a steep increase and a steep decrease before and after reaching its first minimum. This indicates the well-studied explosive growth of α-grains once they successfully nucleate. In contrast, the nano-treated sample's α-peak amplitude is about 20% lower, while the slope after reaching its maximum appears to be less steep. This suggests that the growth of the nano-treated alloy's alpha grains decelerated, as previously hypothesized, resulting in a more continuous nucleation.

Comparing the solid fraction curves derived from the heat flow curve, shown in **4b**, this phenomenon becomes more evident. The sharp turnover point in the pure sample's solid fraction curve is characteristic of the material's unfavorable solidification mechanisms. For the sample welded with filler containing 1.7 vol% TiC, this turnover point is flattened and shows a continuous grain growth up to the crucial terminal stages of solidification, where solidification cracking is initiated. Furthermore, the onset of secondary phase nucleation is delayed by almost 12 °C for the nano-treated alloy. This delay, in combination with pinned and restricted growth of α-grains, support the theory of improved flowability of the liquid melt throughout the melting zone's solidification process.

**Mechanical properties**. To showcase the effect of nano-treated AA7075 filler material on the weld's mechanical properties, microhardness and tensile tests have been performed. To highlight the superior performance of this filler material over the conventional ER5356 filler, cross-sectional microhardness mappings for welds performed with ER5356 and nano-treated AA7075 filler material are shown in Fig. 5a. The high microhardness of the latter's melting zone in as-welded condition, compared to the conventional filler rod, can be attributed to an

altered chemical composition and commonly known strengthening mechanisms in metal matrix nanocomposites, such as the Hall-Petch effect[29] and Orowan strengthening[30]. Since the parameters were kept constant for all welding experiments, it is also noteworthy that degradation of the PMZ and HAZ are mitigated when welding with the nano-treated filler material. An explanation for this can be found in Ma's investigation of laser processed nickel nanocomposites[18]. Even though, in the current study, only the melting zone contains nanoparticles, a manipulation of melt zone viscosity and thermal conductivity is expected to alter the heat flow of the system.

An additional major advantage of nano-treated AA7075 filler material becomes evident when comparing the microhardness results of the two samples after post weld heat treatment (see Methods). Since the nanocomposite filler rod mimics the chemical composition of the base material, the melting zone becomes fully heat treatable and even exceeds the hardness level of the base material due to nanoparticle reinforcement. Furthermore, the pinning effect of the nanoparticles restricts grain growth within the melting zone during the heat treatment, yielding a grain size of $13 \pm 6.2\,\mu m$ after heat treatment. In contrast, with the conventional ER5356 welded as v-groove, only about 20% of the base material's strengthening alloying components are diluted into the melting zone. Therefore, the increase in hardness within the MZ after heat treatment is much less significant.

To further investigate the applicability of this filler material, transversal tensile bars were cut from the weld joint and tested in as-welded and heat-treated condition. Figure 5b shows a selection of welded higher strength aluminum alloys, as well as the results of the present study. For the as-welded case the tensile strength reached a maximum of 392 MPa (average $376 \pm 15$ MPa) at 1.5% elongation, which is considerably higher than the tensile properties reported when using conventional ER5356.

Although the PMZ would have been the designated fracture zone, considering its hardness, the specimen fractured within the melting zone. An improved mechanical performance can be expected when welding defects are reduced and the overall welding parameters are optimized.

As already indicated by the cross-sectional microhardness, the true merit of the nano-treated filler, regarding mechanical properties, becomes evident when heat-treating the tensile bars to T6 condition. Tensile testing of the heat-treated specimen reached 551 MPa (average $531 \pm 19$ MPa) ultimate tensile strength, which is within 93% of AA7075-T6's wrought value[31], at 5.21% elongation. It is remarkable that the specimen exhibiting the highest tensile properties fractured at the onset of the tensile bar's gauge at a great distance from the MZ and PMZ. This indicates that with heat treatment, the resulting fusion joint is possibly as strong as the original wrought material. A summary of transversal ultimate tensile strengths of nano-treated AA7075 welds compared to AA7075 welded with ER5356 filler material and other commonly welded aluminum alloys can be found in Fig. 5b.

## Discussion

In summary, a nano-treated AA7075 filler rod was utilized to disable the driving mechanisms of hot cracking for gas tungsten arc welding of AA7075. Here, the presence of TiC nanoparticles during solidification of the melting zone modified the alloy's α-grain and secondary phase morphologies, yielding a crack free fusion joint. The melting zone's grain morphology was quasi spherical, and the commonly found dendritic grain growth causing solidification cracking was eliminated. Transversal tensile bars cut from the fusion joints showed an extraordinary ultimate

tensile strength of up to 392 MPa in as-welded condition, while reaching up to 551 MPa with post weld heat treatment, which is 96% of the wrought material's property. This method and introduced mechanisms provide an innovative pathway to fabricate filler wires which enable arc welding for other hard to weld material systems and have the potential to improve welding of dissimilar materials.

## Methods

**Filler rod fabrication**. TiC as reinforcement was chosen due to promising incorporation results and satisfactory particle-matrix (aluminum, as well as $Mg(Zn,Cu,Al)_2$ phase) lattice match. Previous research showed that aluminum and TiC can stably coexist above a temperature of 750 °C[32]. Furthermore, the materials have good wettability ($\theta \approx 51°$) above 800 °C[33]. An SEM image of the TiC particles (40–60 nm in diameter) used can be found in Supplementary Fig. 1. Alatorre et al. suggested that a higher content of AA7075 in the melting pool leads to higher as-welded strength and increases the recovery effect of post weld heat treatment[28]. Therefore, to showcase the effectiveness of nanoparticles in preventing hot cracking, the welding filler rods were designed as an AA7075 containing 1.7 vol% percent of TiC.

To incorporate TiC into aluminum, the method of flux assisted liquid state incorporation was chosen due to the laboratory's experience regarding this process and its scalability[34,35]. As flux, $KAlF_4$ was chosen for its promising results in previous research. Aluminum/TiC nanocomposite containing 8 vol% TiC was fabricated as a master. To match the chemical composition of the AA7075 base material, suitable amounts of aluminum (to dilute the master material's TiC content), zinc, copper, magnesium, and chromium were added to fabricate an AA7075 metal matrix nanocomposite containing 1.7 vol% of TiC. The composite was casted into billets and hot extruded to 3.175 mm welding rods. SEM images of a longitudinal welding rod cross-section can be found in Supplementary Figs. 2 and 3.

Energy dispersive X-ray spectroscopy (EDS) analysis of ion-milled rod segments showed the alloying components were within reasonable proximity of AA7075's chemical composition, shown in Table 1.

**Welding procedure**. Welding experiments were conducted by fusing $152.4 \times 76.2 \times 3.175$ mm AA7075 sheets as butt weld with v-groove, clamped onto a copper backing plate. As a reference, conventional ER5356 and the novel AA7075 MMNC filler rod, 3.175 mm in diameter, were welded under equal parameters for power source (Lincoln Aspect 375) and welding robot, shown in Table 2.

**Sample preparation**. The fused AA7075 base plates were cut into transversal tensile bars using an AgieCharmilles CUT 200 wire EDM machine. The residual slugs were first ground by 400, 600, 800, and 1200 grinding paper and finally polished using 1 μm $Al_2O_3$ compound for optical microscopy, SEM, and Vicker's Microhardness characterization. SEM samples were further ion milled. To enhance the visibility of grains and contrast when using polarized light, the OM samples were additionally etched with Weck's reagent. For TEM characterization, a sample of approximately 46 nm thickness was cut from the MZ using Focused Ion Beam (FIB) and characterized with a Titan S/TEM (FEI) at 300 kV.

### Table 1 Chemical composition of fabricated AA7075 + 1 vol % TiC filler material

|  | Mg | Cu | Zn | Cr | Al |
|---|---|---|---|---|---|
| AA7075 | 2.1–2.9 | 1.2–2 | 5.1–6.1 | 0.18–0.28 | bal |
| Nano-treated AA7075 | 3.2 | 1.2 | 6.4 | 0.15 | bal |

### Table 2 Welding parameters

| Parameter | Value |
|---|---|
| Type of current | Constant current |
| Current | 180 A |
| Output frequency | 180 Hz |
| Balance | 85% |
| Argon flow rate | 18 cc/min |
| Electrode gap | 2 mm |
| Electrode forward speed | 60 mm/min |
| Start delay | 1.5 s |

**Post weld heat treatment (PWHT)**. Since no research is available on the effect of nanoparticles on the solutionizing and aging temperatures/times, a procedure proven to be adequate for post weld heat treatment was chosen. The samples were solutionized at 480 °C for 1 h and then water quenched at 25 °C. In a second step, the samples were aged at 120 °C for 19 h and then cooled to room temperature in air.

**Vicker's microhardness**. Using polished, transversal vertical cross-sections, the microhardness tests were conducted at half of the base material's thickness using machine settings of 200 g force and 10 s dwell time.

**Tensile testing**. The fused AA7075 base material plates were cut to tensile specimen (ASTM–E8 with reduced grip length) using a Georg Fisher AG cut 200 Wire EDM machine and were further polished. Tensile tests were conducted using a 100 kN load cell and a test speed of 1.27 mm per minute. UTS elongation was measured manually after removing specimen from the tensile testing machine.

**DSC**. 44.63 mg and 43.02 mg samples were cut from the melting zones welded with nano-treated and pure AA7075 welding rod, respectively, and investigated using an Elmer Perkins DSC 8000. In alumina crucibles, the samples were heated from 25 °C to 670 °C at 350 °C/min, held at 670 °C for 10 min, and then cooled to 350 °C at 10 °C/min in order to observe the samples' heat flow within the semi solid zone. The baseline, obtained by running the program with alumina crucible only was subtracted, and the resulting data was corrected for the mass difference.

## Data availability

The datasets generated during and/or analyzed during the current study are available from the corresponding author on reasonable request.

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

## Acknowledgements

We thank Moheimin Khan for his support in fabricating nano-treated welding rods. We also thank Travis Widick for his advice and insightful discussions regarding the welding process and Ting-Chiang Lin for supplementary DSC experiments.

## Author contributions

X.C.L. and M.S. conceived the idea and designed the experiments. M.S. fabricated the AA7075 + 1.7 vol% TiC welding rods and conducted the welding experiments. M.S. and C.C. characterized micro/nano structures and mechanical properties. S.P. conducted DSC analyses and analyzed the data. M.S. and X.C.L. wrote the manuscript. X.C.L. supervised the whole work.

## Additional information

**Competing interests:** The authors declare no competing interests.

