## [Peer Review File · Nature Communications]

Reviewers' comments:

Reviewer #1 (Remarks to the Author):

This paper reports a novel approach to address the long-standing challenge in arc welding of 7075 alloy, an alloy of high demand in aerospace and automotive industry. The nanoparticle approach reported here solves the three challenges associated with welding of 7075 simultaneously: solidification cracking, liquation cracking, and weakening of weldment by filler material. It opens up a new avenue for solving welding problems in many unweldable materials.

Nanoparticles play the key role in solving the problem. However, the size, size distribution of the nanoparticles used, the distribution of nanoparticles in the welding wire and in the as-solidified materials are not presented in the paper.

Various nanoparticles are available from market. Why TiC was selected? A discussion about the criteria to select nanoparticles will be important for guiding the design of nanoparticle-enabled filler wires for other unweldable materials.

Reviewer #2 (Remarks to the Author):

The 7075 high-strength aluminum alloys demonstrate the hot-crack susceptibility during fusion welding, which is therefore regarded as one category of hard-to-weld aluminum alloy. In the present work, the authors have shown that the 7075 aluminum alloy can be safely arc welded without hot cracks by using the filler wire containing TiC nanoparticles. The addition of TiC nanoparticles enables the phase control during welding. The obtained joints exhibited fine globular grains and a modified secondary phase, due to the addition of nanoparticles, which contributed to the improvement of tensile strengths of the joints. The current work contains some useful elements that may solve the crack-formation problem during welding of 7075 hard-to-weld aluminum alloy. However, after a careful reading of the present article, I have the following questions and comments on the work.

/1/ The title of this article is a bit exaggerated, and it is necessary to accurately address the research contents of this work. The 7075 aluminum alloy is hard-to-weld, but it is definitely not "unweldable". Furthermore, "nanotechnology-enabled phase control" is actually the addition of TiC nanoparticles to the 7075 filler wire. This is a method of preparing metal-based nanocomposites, but it seems to be far-fetched to raise it to the height of nanotechnology.

/2/ When I first read this article, I found that the key research idea of this work is very similar to a recent paper entitled "3D printing of high-strength aluminium alloys" published in Nature (doi:10.1038/nature23894); both papers belong to the materials processing/engineering research field of difficult-to-process aluminum alloys. The key idea of this study, i.e., to improve the mechanical properties by adding nanosized ceramic particles to adjust the solidification behavior and crystalline structure of matrix metal, is identical to the above-mentioned Nature paper. This obviously affects the innovation of the present work.

/3/ As a paper submitted for the consideration in the very high-level journal Nature Communications, I am sorry but I have to say that this article only contains the very rough micro-structural characterization, which is not enough for a paper on materials science. Only the very basic optical microscope and scanning electron microscope (SEM) characterizations have been carried out in the present work; more other methods of microstructural analysis, such as Transmission Electron Microscope TEM and Electron Backscattered Diffraction EBSD, etc, are needed to accurately describe phase constitutions and micro-structural features of the joints. It is necessary to disclose, from the atomic scale, how the nanosized TiC ceramic particles affect the solidification behavior of the Al-matrix metal and how the atomic interfacial structure is formed

and evolved.

Reviewer #3 (Remarks to the Author):

This manuscript reports an experimental study focused on demonstrating that arc welding of hard-to-weld AA7075 can be successfully achieved by adding TiC nano-particles to a AA7075 filler using a fabrication method developed by the authors. This is a significant achievement, in practical terms, as it is well-known that sound welds cannot normally be obtained through fusion welding for this alloy.

This study is therefore clearly of interest to the research community at large in demonstrating the potential benefits of nano-particles in controlling hot tearing but also solidified microstructures from which mechanical properties depend, and with significant potential impact as far as industrial applications are concerned. Results are novel and important enough to justify publication in Nature Communications.

The paper is well-written but it would benefit from addressing the following comments:

1. It is not clear why the 1.7% concentration of TiC nano-particles has been selected. Have the authors carried out a study demonstrating an optimal behaviour for this particular composition? If so, results should be added.
2. How many tests have been carried out? Statistical significance of the results should be demonstrated.
3. Planes orientation and locations of microstructural observations (Fig. 2 and Fig. 3) and micro-hardness measurements (Fig. 5) with respect to the welded plate geometry should be clarified using, for instance, sketches and/or close-up views of lower magnification images.
4. Page 5. What do the Authors mean by "error pattern" as in Fig. 1a? I guess they are simply referring to the crack shown in Fig. 1b.
5. Page 14. "..., the PMZ would have been the designated fracture zone, ..." is repeated twice in the same sentence.
6. All stress values throughout the paper should be rounded without the need for decimals.

RESPONSE TO REFEREES' COMMENTS

We appreciate the insights from the reviewers and especially are encouraged by the fact that the reviewers saw merit in our work. The comments certainly inspired us to better present the novelty of our work. Detailed responses to specific comments from the referees are listed as follows.

Response to Referee #1:

Comments:

A. Nanoparticles play the key role in solving the problem. However, the size, size distribution of the nanoparticles used, the distribution of nanoparticles in the welding wire and in the as-solidified materials are not presented in the paper.

Response: Thanks for the comments. The nanoparticles used in this study are commercially available from US Research Nanomaterials have a size of 40-60 nm. We added one SEM image of the as received particles to the supplementary material. The methods section article was supplemented accordingly:

“TiC particles of 40-60 nm in diameter acquired from US Research Nanomaterials, Inc, shown in the supplementary material were used.”

Two figures were added to the supplementary material to show the particle distribution of the extruded welding rod at low and high magnification. The methods section article was supplemented accordingly:

“The nanocomposite was casted into billets and hot extruded to 3.175 mm welding rods. SEM images of a welding rod cross-section, revealing its particle distribution and dispersion, can be found in the supplementary material”

The particle dispersion of the as solidified melting zone can be found in Figure 3e and especially Figure 3f. It is shown, that the nanoparticles are mainly present in the alloy's secondary phase while controlling the growth of the alpha aluminum grains.

B. Various nanoparticles are available from market. Why TiC was selected? A discussion about the criteria to select nanoparticles will be important for guiding the design of nanoparticle-enabled filler wires for other unweldable materials.

Response:

Thank you for the comment. We added additional information and literature to point out the favorability of the aluminum-TiC system

“TiC as reinforcement was chosen due to promising incorporation results and the mentioned satisfactory particle-matrix (Aluminum as well as Mg(Zn,Cu,Al)₂ phase) lattice match. Previous research showed that aluminum and TiC can stably coexist above a temperature of 750 °C. Furthermore, the materials have good wettability ($\theta \approx 51^\circ$) above 800 °C.”

Response to Referee #2:

Comments:

- A. The title of this article is a bit exaggerated, and it is necessary to accurately address the research contents of this work. The 7075 aluminum alloy is hard-to-weld, but it is definitely not “unweldable”. Furthermore, “nanotechnology-enabled phase control” is actually the addition of TiC nanoparticles to the 7075 filler wire. This is a method of preparing metal-based nanocomposites, but it seems to be far-fetched to raise it to the height of nanotechnology.

Response: Thanks for the comments. We will change the title to “Nanoparticle-enabled phase control”. In the meantime, we would appreciate if the reviewer can agree that the AA7075 is generally considered as not weldable by arc welding process, which is stated by leading welding companies such as Lincoln Electric and ESAB. Please see:
<http://www.esabna.com/us/en/education/blog/how-do-i-weld-2024-and-7075.cfm>
<https://www.lincolnelectric.com/en-us/support/welding-solutions/Pages/aluminum-faqs-detail.aspx#question8>

- B. When I first read this article, I found that the key research idea of this work is very similar to a recent paper entitled “3D printing of high-strength aluminium alloys” published in Nature (doi:10.1038/nature23894); both papers belong to the materials processing/engineering research field of difficult-to-process aluminum alloys. The key idea of this study, i.e., to improve the mechanical properties by adding nanosized ceramic particles to adjust the solidification behavior and crystalline structure of matrix metal, is identical to the above-mentioned Nature paper. This obviously affects the innovation of the present work.

Response: Thanks for the constructive comments. However, we respectfully disagree with the comments on the novelty and originality of this paper. In the aforementioned Nature paper, only AA7075 powder decorated with hydrogen-stabilized zirconium nanoparticles, serving as grain refiner, was processed by laser. These particles dissolve in the melt pool and enhance the nucleation of the solidifying material. We believe that with this grain refining approach, the very high solidification rate of Selective Laser Melting (SLM) is necessary to yield a crack free specimen. However, when introducing ceramic nanoparticles to the melt pool, which remain as solid particulates throughout the solidification process, not only nucleation is enhanced, but also the propagation of the solidification fronts is phase controlled. It is therefore possible to achieve non-dendritic, globular grains, which is crucial to inhibit solidification cracking, at the considerably lower solidification rate of TIG welding.

We respectfully acknowledge that in the paper “3D printing of high-strength aluminium alloys” AA7075 powders decorated with ceramic nanoparticles were presented and the authors proposed that processing these materials would also yield crack-free specimen. However, this hypothesis was not substantiated with experimental results. Therefore, this paper shows for the first time that the addition of ceramic nanoparticles can impede solidification cracking in AA7075 by controlling the growth of the solidifying grain without the necessity of typical high solidification rates from SLM.

C. As a paper submitted for the consideration in the very high-level journal Nature Communications, I am sorry but I have to say that this article only contains the very rough micro-structural characterization, which is not enough for a paper on materials science. Only the very basic optical microscope and scanning electron microscope (SEM) characterizations have been carried out in the present work; more other methods of microstructural analysis, such as Transmission Electron Microscope TEM and Electron Backscattered Diffraction EBSD, etc, are needed to accurately describe phase constitutions and micro-structural features of the joints. It is necessary to disclose, from the atomic scale, how the nanosized TiC ceramic particles affect the solidification behavior of the Al-matrix metal and how the atomic interfacial structure is formed and evolved.

Response: Thanks for the constructive suggestions. To reveal the phase constitutions, micro-structural features and interfaces at atomic scale, we further used high resolution TEM at the secondary phase with TiC nanoparticles. By analyzing the crystal structures, we readily identify TiC nanoparticle and MgZn₂ secondary phase orientated to [01 $\bar{1}$] and [0001] zone axis, respectively. A crystal orientation relationship of (111)_{TiC} // (11 $\bar{2}$ 0)_{MgZn₂} was identified. The misfit at the TiC-MgZn₂ interface is calculated to be approximately 5.6% which indicates a clean semi-coherent interface. The good lattice matching indicates that TiC nanoparticles could be a potent inoculation for nucleation of MgZn₂ secondary phase and also explains why TiC nanoparticles tend to affiliate to the secondary MgZn₂ phase and effectively modify MgZn₂ phase for improved welding quality. Since the observed grain sizes are well above 1 micron, we considered color etching and optical microscopy a simple but exhaustive way to investigate the weld beads grain morphology. We contemplated, adding EBSD analysis to our paper before submitting the first draft but could not see its additional value for the investigated phenomenon. We added the following to the revised manuscript:

“To clearly reveal the interface between the TiC nanoparticles and secondary Mg(Zn,Cu,Al)₂ phases, TEM analysis at atomic scale was utilized. Figure 3f shows the SEM image of the TEM sample cut from the nano-treated eutectic Mg(Zn,Cu,Al)₂ areas with TiC nanoparticles. As marked in the figure, it is observed that most TiC nanoparticles stay either inside secondary phase or at the boundary between Al matrix and Mg(Zn,Cu,Al)₂ phases. This indicates that TiC nanoparticles prefer to stay with Mg(Zn,Cu,Al)₂ phases during solidification and effectively modify the size, shape and distribution of secondary phases in MZ. Figure 3g shows the typical interface between TiC nanoparticles and secondary phase. Figure 3h is the Fourier-filtered atomic resolution TEM image corresponding to Figure 3g. This secondary phase is identified as MgZn₂ phase by the atomic structure. As marked in Figure 3h, (11 $\bar{2}$ 0) and (1 $\bar{2}$ 10) planes of MgZn₂ were identified with an angle of 120°. The MgZn₂ phase is orientated to [0001] zone axis. Meanwhile, in TiC nanoparticles, (111), ($\bar{1}$ 11) and (200) planes are identified and marked in the atomic structure. This TiC nanoparticle is orientated to [01 $\bar{1}$] zone axis. The (111) planes of TiC are shown to be parallel with (11 $\bar{2}$ 0) planes of MgZn₂. The ($\bar{1}$ 11) planes of TiC have an angle of approximately 10° between (1 $\bar{2}$ 10). The plane distance of ($\bar{1}$ 11) TiC and (1 $\bar{2}$ 10) MgZn₂ are 0.2499 nm and 0.2609 nm, respectively. Thus the misfit at the TiC-MgZn₂ interface is calculated to be approximately 5.6% which indicates a semi-coherent interface. The good lattice matching also explains why TiC nanoparticles tend to affiliate to the secondary MgZn₂ phase and effectively modify MgZn₂ for improved welding quality.”

Response to Referee #3:

Comments:

A. It is not clear why the 1.7% concentration of TiC nano-particles has been selected. Have the authors carried out a study demonstrating an optimal behaviour for this particular composition? If so, results should be added.

Response: Thanks for the comments. Since the ~7 vol% Al-TiC master we used to fabricate the welding wire exhibited a concentration gradient of nanoparticles it was difficult to design an exact vol% in the resulting AA7075+TiC welding rod in this study. The welding rod containing 1.7 vol% was initially designed for 1.5 vol%. We also did rudimental investigations for 3.5 vol% which showed an almost identical grain morphology (Grain size $7.3\pm 3.6\ \mu\text{m}$, Figure 1), as well as increased micro hardness in MZ and HAZ (Table 1). Since the goal of our paper was not to increase the weld's properties by forming a high concentration nanoparticles in the metal matrix nanocomposite, but to showcase the effectiveness of nanoparticle-induced phase control, we decided to pursue the low volume percentage approach and conduct a more detailed study.

Figure 1: Microscope image of MZ boundary welded with AA7075 + 3.5 vol% TiC

Distance from fusion interface (mm)	-1 (MZ)	0 (Fusion line)	0.1 (PMZ)	2 (HAZ)	4.5 (HAZ)	6 (HAZ)
5356	98.6	102	115.8	118.2	122.6	132.1
7075 + 1.7v%TiC (HV)	131	133	117	124	135	141
7075 + 3.5v%TiC (HV)	143	135	120	129	139	146

Table 1: Microhardness of AA7075 + 3.5 vol TiC

B. How many tests have been carried out? Statistical significance of the results should be demonstrated.

Response:

We typically performed 4 welds for the 1.7 vol% TiC filler wire in each trail, while none of them cracked. We typically cut 6 tensile bars. Three of them were tested as welded, the other three were tested after T6 heat treatment. Error bars were added to Figure 5.

Hardness testing was performed for 4 specimens. Hardness values were obtained in 0.5 mm steps with 3 measurements per step.

C. Planes orientation and locations of microstructural observations (Fig. 2 and Fig. 3) and microhardness measurements (Fig. 5) with respect to the welded plate geometry should be clarified using, for instance, sketches and/or close-up views of lower magnification images.

Response:

Thanks for the remark. We used standard terminology to characterize weld beads.

Figure 2: Center line of vertical transversal cross-section for microhardness testing

Figure 3: Horizontal cross-section

However, we agree that these definitions could be more descriptive and changed parts of Fig 2's caption.

“Optical microscope characterization of horizontal weld bead cross-sections, in-plane with the welded base plates, as shown in Figure 1”

D. Page 5. What do the Authors mean by “error pattern” as in Fig. 1a? I guess they are simply referring to the crack shown in Fig. 1b.

Page 14. “..., the PMZ would have been the designated fracture zone, ...” is repeated twice in the same sentence.

Response:

Thanks for the comments. We were indeed referring to the crack shown in Fig 1b. We corrected both mistakes.

F. All stress values throughout the paper should be rounded without the need for decimals.

Response: Thanks for the comments. We revised the paper accordingly.

References

- [1] Contreras, A.; León, C.A; Drew, R.A.L; Bedolla, E. (2003): Wettability and spreading kinetics of Al and Mg on TiC. In Scripta Materialia 48 (12), pp. 1625–1630. DOI: 10.1016/S1359-6462(03)00137-4.
- [2] Arpón, R.; Narciso, J.; Louis, E.; García-Cordovilla, C. (2013): Interfacial reactions in Al/TiC particulate composites produced by pressure infiltration. In Materials Science and Technology 19 (9), pp. 1225–1230. DOI: 10.1179/026708303225004459.

REVIEWERS' COMMENTS:

Reviewer #1 (Remarks to the Author):

The authors addressed all my concerns. I recommend the publication of the paper.

Reviewer #2 (Remarks to the Author):

The authors have taken efforts to further revise the manuscript, addressing the points sufficiently in the first-round review comments. The current version of the manuscript is basically acceptable for publication. However, the following point needs to be further addressed.

For the headings of different sections in this manuscript, the authors used the different microstructural and property analysis methods, for instance, Optical characterization, Characterization by Scanning Electron Microscope (SEM) and Transmission Electron Microscope (TEM), Differential Scanning Calorimetry (DSC) Study, etc. the authors are suggested to disclose the main scientific issues in the headings, rather than using the different experimental methods for description.

Reviewer #3 (Remarks to the Author):

I am happy with the additions made to the paper in the revised version and responses provided by the authors. I therefore recommend publication.

SECOND RESPONSE TO REFEREES' COMMENTS

Response to Referee #2:

Comments:

- A. The authors have taken efforts to further revise the manuscript, addressing the points sufficiently in the first-round review comments. The current version of the manuscript is basically acceptable for publication. However, the following point needs to be further addressed.

For the headings of different sections in this manuscript, the authors used the different microstructural and property analysis methods, for instance, Optical characterization, Characterization by Scanning Electron Microscope (SEM) and Transmission Electron Microscope (TEM), Differential Scanning Calorimetry (DSC) Study, etc. the authors are suggested to disclose the main scientific issues in the headings, rather than using the different experimental methods for description.

Response:

Thanks for the comment. We changed the subheadings to reflect the major findings or scientific issues addressed in the following chapters.